# Ensuring Security and Energy Efficiency of Wireless Sensor Network by Using Blockchain

**Abdul Rehman** [1], **Saima Abdullah** [1,2], **Muqaddas Fatima** [2], **Muhammad Waseem Iqbal** [3], **Khalid Ali Almarhabi** [4], **M. Usman Ashraf** [5,*] and **Saqib Ali** [6]

1. The Department of Computer Science and Information Technology, University of Science and Technology of China, Hefei 230026, China
2. The Department of Computer Science, The Islamia University of Bahawalpur, Bahawalpur 63100, Pakistan
3. Faculty of Computer Science and Information Technology, The Superior University, Lahore 54600, Pakistan
4. DDepartment of Computer Science, College of Computing in Al-Qunfudah, Umm Al-Qura University, Makkah 24381, Saudi Arabia
5. Department of Computer Science, GC Women University, Sialkot 51040, Pakistan
6. Department of Computer Science, University of Agriculture Faisalabad, Faisalabad 38000, Pakistan
* Correspondence: usman.ashraf@gcwus.edu.pk

**Abstract:** With the advancement of new technology, security is the biggest issue nowadays. To solve security problems, blockchain technology will be used. In recent work, most of the work has been done on homogeneous systems, but in our research, the primary focus is on the security of wireless sensor networks using blockchain. Over the last few decades, the Internet of Things (IoT) has been the most advancing technology due to the number of intelligent devices and associated technologies that have rapidly grown in every field of the world, such as smart cities, education, agriculture, banking, healthcare, etc. Many of the applications are developing by using IoT technologies for real-time monitoring. Because of storage capacity or low processing power, smart devices or gadgets are vulnerable to attack as existing cryptography techniques or security are insufficient. In this research work, firstly, we review and identify the privacy and security issues in the IoT system. Secondly, there is a solution for the security issues, which is resolved by blockchain technology. We will check the wireless sensor network to see how data work on distributed or decentralized network architecture. Wireless sensor network clustering technique was introduced by researchers for network efficiency because when the workload spreads, the system will work faster and more efficiently. A cluster comprises a number of nodes, and the cluster head manages the local interactions between the nodes in the cluster (CH). In general, cluster members connect with the cluster head, and the cluster head aggregates and fuses the data acquired in order to save energy. Before approaching the sink, the cluster heads may additionally create another layer of clusters among themselves. The clustering concept divides data traffic into several groups similar to the other data points in the same data point. In contrast, this data point is dissimilar from other data points in another group. All results are presented at the end of this study paper, in which we will see the network or nodes' performance in the specific area of the network, how it works, and how efficient it is. Likewise, Blockchain also works in a distributed manner.

**Keywords:** Internet of Things; blockchain; wireless sensor network; security; privacy; consensus mechanism



## 1. Introduction

The "Internet of Things" is the rising field in the technological era. IoT is currently used in various applications such as healthcare, food packaging, structural monitoring, groundwater remediation, accident avoidance, wildlife monitoring, home automation system, and emergency response in industries. The Internet of Things is a collection of many little objects and low-cost battery-powered sensor nodes commonly used to

monitor the distributed environment. The failure of any sensor node can badly affect the performance of the whole IoT network. As everyone knows, the IoT (Internet of Things) works with less human interaction, and there is no limit for location and environment; this raises some security, energy, and recovery issues.

Blockchain technology will be used for the security problem in the proposed system, where systems ensure the network's security. Confidentiality in wireless sensor networks is made with hundreds and thousands of small appliances that are fit for detecting and preparing and have some correspondence capacities. The security components offered in regular organizations are, for all intents and purposes, not appropriate to be executed in WSNs due to their asset-compelled nature. These organizations cannot deal with higher overhead bits. For increased security, we use blockchain technology, and some protocols are used for this. The most popular protocol is the LEACH, a homogenous protocol used for the same type of transaction. The motivation of this work is to secure the WSNs, and there are some challenges related to the security of WSNs.

As it is known that WSNs are open-access networks, anyone can operate them. Most of the protocols used in WSNs are insufficient for the security method. It is challenging to employ strong protection on sensors due to their complexity. Blockchain innovation has arrived at a point of dissatisfaction, particularly concerning sharing economy stages. Some essential security goals are "confidentiality, integrity, and availability," as everyone knows all the communication devices connected through the Internet. IoT security plays an important role; there are three layers of IoT security. The first layer is the physical, network, and application layer attack. The privacy and security issues still need to be addressed. As the "IoT" devices are resource limitation devices, current security protocols or mechanisms are not appropriate. Inconsequential algorithms (cryptographic algorithms) or communications protocols are much needed for "IoT" devices [1].

There is one problem that exists in the Blockchain, which is the Byzantine Generals Problem. There are many algorithms used to solve this problem, such as Proof-of-Work, Proof-of-stake, Delegated-Proof-of-Stake, Proof-of-Authority, Proof-of-Capacity/Space, Proof-of-weight, Proof-of-Importance, Proof-of-Burn, Proof-of-Activity, Practical-Byzantine-Fault-Tolerance, Federated-Byzantine-Agreement, Delegated-Byzantine-Fault-Tolerance, and Directed-Acyclic-Graphs. As mentioned above, all algorithms have some properties and pros and cons according to their capability. Every algorithm is used for a different issue because all have other qualities. Nowadays, the Internet of Things (IoT) is the most advanced strategy, which is an arrangement of interrelated smart devices, items or objects, power-driven and electronic machines, beings, or individuals that will be used to give unique identities such as IP addresses and the energy or capacity to exchange data or important information through a network without relying on human-to-human or human-to-PC collaboration.

With the advancement of information technology, gadgets or smart devices such as tablets, laptops, phones, and others which communicate wirelessly have become essential needs in everyone's life. Many of the applications are developed by using IoT technologies for monitoring in real-time. Because of storage capacity or low processing power, smart devices or gadgets are vulnerable to attack as existing cryptography techniques or security are insufficient. In this research, firstly, we review and identify the privacy and security issues in the IoT system. Secondly, there is a solution for the security issues, which is resolved by blockchain technology [2].

Our critical fear is that security attacks are becoming more advanced and high-grade. With six principles, autonomy, immutability, anonymity, transparency, and open sources, blockchain innovation has emerged as one of the most significant emerging concepts over the last several decades.

As everyone knows that communication is done wirelessly, there is a complete sensor network known as Wireless Sensor Network. This is the most critical technology, which consists of low-power, low and high operating nodes. WSN consists of sensors called nodes. These nodes are essential and perform many tasks used to collect the data. The

collecting data node is known as the source node, and there is one sink node that contains data from all source nodes. The sink node is more potent than other nodes because it has high computing power. A wireless sensor network measures how well the sensor nodes can observe the data. Connectivity is vital in WSN. A flow diagram of security process is presented in Figure 1 as follows.

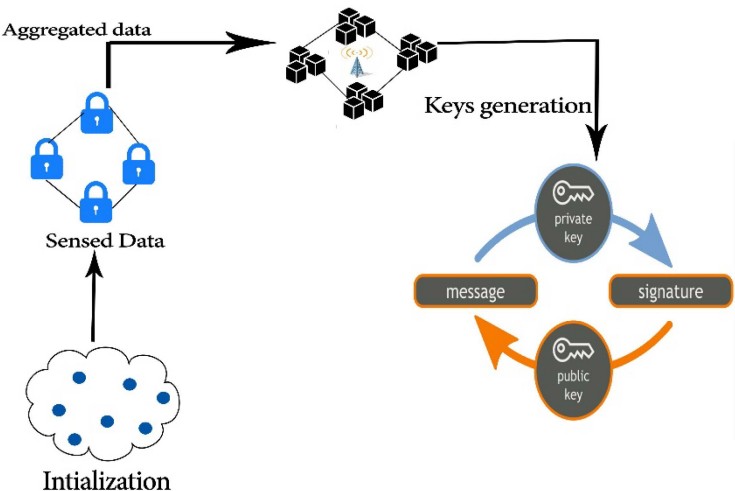

**Figure 1.** Flow diagram of the security process.

Our Contribution: This study makes many significant contributions, including introducing blockchain technology, which is used to increase the security of wireless sensor networks. In IoT applications, layer-by-layer privacy and security concerns are first identified. How the IoT is integrated with blockchain technology is then explained, and its ability to meet privacy and security concerns is analyzed. Security-related blockchain technology is recognized and thoroughly addressed. Wireless sensor networks use blockchain technology to authenticate the data in a decentralized or distributed system. The most important contributions of the proposed research work are: introducing blockchain technology to strengthen the security of wireless sensor networks and ensure the network's efficiency.

## 2. Privacy and Security Issues

IoT-based innovative applications took the place of traditional applications with the advancement of IoT technology. Many jobs have been undertaken concerning IoT structure and protocols of IoT-based products. The privacy and security problems still need to be focused on IoT approaches to face security and privacy concerns, as discussed in [3]. The device has limitations, and a new attack paradigm for IoT-based applications is outlined layer-wise. The gadget has limits, and a further assault paradigm for IoT-based applications is outlined layer-wise. IoT apps are built using a structure described in [4], where the inventors identify eight possible frameworks for constructing applications and their security and privacy challenges.

Security and privacy issues, such as authentication and data protection, are the most challenging aspects of designing IoT applications, according to [5,6], where the authors show how blockchain, cloud computing, and device understanding might be used to solve the problem. There are three layers to the IoT infrastructure: physical, network, and application. Each of these tiers has a security vulnerability.

### Security Challenges in IoT

The security problems of IoT applications are discussed in this section. Most IoT applications deal with the physical, network, and application layers that make up a three-layer architecture level of application. The gateway connects the components in the physical layer. The capabilities of the physical gadget are restricted, and it is vulnerable to an attacker.

It is not recommended to change the overall situation if a physical element is compromised. The thief was the one who stole it. When each layer is available, the system must deal with the issue of cybersecurity.

**Node capture attacks**: IoT applications place intelligent devices in various locations to connect to the network, making it possible for attackers to grab or replace them with an unsuitable device. It is hard to distinguish between the brilliance and the false nodes in this kind of physical assault. The attacker may obtain crucial details about the software through this attack. The network has to be secured against attacks of this kind.

**Replay attacks:** The perpetrators broadcast identical information to the network after intercepting it on telecommunications. An attacker may seize control of intelligent gadgets in an "IoT infrastructure" and transmit data as if they were a legal network node.

Attacks from the side channel: In side-channel attacks, the attacker tries to decipher the ciphertext to retrieve the plaintext. Gaining the key through a time constant is crucial since most encryption methods employ key exchanges for encryption and decryption.

**False data injection:** To read environmental data, smart devices or sensors are deployed in various places. Sensors and intelligent devices can gather and transmit data to the following layer. Smart gadgets are susceptible to attackers due to their resource restrictions. The attacker tries to intercept the gadget or extract the value from an unprotected communication channel to introduce fake data into the network.

**Eavesdropping:** An eavesdropping attack takes place in an IoT environment where compromised smart devices are reading one another's messages because the communication link is not protected. In a passive assault, the advertiser collects information from an unreliable communication route.

**Spoofing:** The attacker attempts to obtain access to the network layer of smart devices, then, after doing so, pretends to be a network node and sends false signals across the network.

**Denial-of-service (DoS) attacks:** Cyber-attacker uses the system's permission to send a large amount of data to a server in an attempt to overload it; as a result, it will be unable to use the entire amount of bandwidth and resources available, costing the victim a significant amount of money.

**Phishing attacks:** With the widespread adoption of the Internet of Things, where each user has a unique identity and utilizes the network interface to access or manage smart devices, hackers may attempt to obtain critical information about savvy users or devices by sending a fake message or email. Users exchange personal information with the network as an IoT-based application to monitor and manage the environment in real time. Therefore, trust management concerns will arise among the network's nodes if a node in the network behaves maliciously.

**Authentication:** Intelligent devices, sensors, actuators, and some smart devices make up an IoT application, which monitors and computes. Smart devices collect data or information, which is then transferred to the next layer for processing and computation. The network node triggers the related event once the calculation is complete.

**Malicious attack:** Smart devices used in IoT applications are exposed to the outside world due to unprotected wireless connections and communication pathways. An attacker might compromise the device by injecting malicious code into the device via the software.

**Unauthorized attack:** Because most IoT devices are connected through a separate router, the attacker uses several techniques to obtain authorization credentials. After obtaining personal credentials, the attacker has access to the network data.

Policy enforcement: One of the most critical security issues in IoT applications is policy, which allows users to utilize smart devices. Adequate rules must be defined by the application's needs to ensure user privacy.

## 3. Related Work

A peer-to-peer network structure to accomplish decentralization is an idiographic feature of blockchain-based technology. The first blockchain technology has improved

the decentralized network structure, which was based on peer-to-peer networking [7,8]. Several current studies employ the topic of application for its relationship with software programming. Software is a programming application that helps you define a goal. Millions of software programs are used, most of which employ a centralized server–client architecture. There are a few decentralized network architectures, even though several network types are dispersed [9,10].

The number of IoT devices and use cases has increased dramatically during the previous decade. Due to their resource limitations, smart devices are susceptible to several types of attacks. A specific point of collapse is one of the worst issues with centralized infrastructure. Data transport and processing provide different security problems for applications at each layer. To address security and privacy issues, academics have lately turned to blockchain as a decentralized solution. The research that has been done to solve issues including trust management, safe storage, confirmation, privacy protection, and access control is compiled in Table 1. According to an assessment of the research, blockchain technology may be able to address some of the security and privacy issues associated with IoT. This study investigated how blockchain technology may be utilized to address several security and privacy issues.

Security on the blockchain is a critical issue that must be considered throughout the system's lifecycle, from requirement analysis to coding and maintenance [11]. On the other hand, scholars in the field of information systems appear to be unaware of the contributions made by researchers from different disciplines.

Today's most common network software application model, the centralized system, directly controls each unit and manages signal processing for each center.

The administration of rights by the central point is completely dependent on individual nodes, and the whole network infrastructure receives and delivers data in accordance with the rights. A distributed network system is known as a peer-to-peer (P2P) network. Torrent file downloading is just one example of the P2P information networks that many online file exchanging and live streaming services use.

Following BitTorrent into operation, blockchain likewise uses the peer-to-peer network protocol. This network's protocols nodes all have the same status and are not part of a centralized control structure or a middleman for transactions. Nodes may join or depart the network at any moment while concurrently providing some or all services. Each network node functions as both a server and a client. The system's processing power, data security, and damage resistance all increase as the number of nodes increases. The well-known technology Bitcoin also uses the P2P network protocol. Trusted central economic organizations operate as a middleman, and each operation carried out by this centralized association is documented and monitored. This involves both the client and the server. The "peer-to-peer" network protocol is used by "Bitcoin," which enables transactions to happen directly between users without needing a middleman [12,13].

A complete blockchain system consists of data blocks for storing data and cryptographic signatures, system logs, peer-to-peer network infrastructure, maintenance system methodologies, workloads for data mining, proof rules, unknown transmission data mechanisms, "Unspent Transaction Output" (UTXO), Merkle trees, and other related engineering concepts. Using these advancements and a steady stream of blockchain networks, the blockchain offers an endless engine on a non-centralized network for services like transmission, verification, and membership [14].

This method may produce a sensor data record from a blockchain transaction history. The blockchain network generates a new block, which generally has a header and a body, every ten minutes. The current block number, beginning block hash value, current timestamp, random value (nonce), current block hash value, and Merkle tree are all included in the block header. The unit content is essentially where the sensor data are found. Every piece of sensor datum is permanently stored in the research system's records block and accessible to the operator. The Merkle tree in the block digitally signs each component of the sensor data acquired, guaranteeing that none of the sensor data obtained are repli-

cated. Once the system has collected all sensor data, the Merkle-tree hash method creates "Merkle-root" values on the block's description section. [15].

Ref. [16] proposed data security procedures in a wireless sensor network scenario.

More study of security attacks and mitigation for wireless sensor networks used for monitoring was advocated in the proposed [17]. In [18], sensor fusion is used in wireless sensor networks for surveillance to identify mobile intruders widely. Ref. [19] proposed a fusion-based system for remote sensing applications based on wireless interactive media sensing devices.

The degree of secrecy provided by blockchain technology is its most enticing feature, yet this might lead to some transparency issues. Agreements self-audit the digitized value ecosystems that manage transactions frequently (e.g., every 10 min). Transparency and corruptions absence are the outcomes of this procedure. It is impossible to associate a specific user with a set of public addresses in a blockchain since the user's identity is concealed behind a sophisticated cipher [20]. There are different security-related works in various fields mentioned below with references.

Proposed research [21] has addressed the creation of a blockchain network for cross-domain image sharing that uses a consensus blockchain to allow patients medical and radiological pictures to be shared. The author sought consensus among a few trustworthy institutions to maintain a more thorough consensus in which the sophisticated security and privacy module could be managed with less effort.

According to proposed research [22], blockchain technology has enhanced medical record transfers in health coverage 4.0 applications. Compatibility of healthcare databases, access to clinical documentation, prescription databases, and device tracking are just a few of the primary advantages of using blockchain in healthcare. In addition, the authors developed a policy for access control techniques to optimize access to medical information across healthcare providers.

S2HS is a new approach for delivering intrinsic confidentiality and reliability to a smart public health system [23]. This study employs two-level blockchain techniques for both internal and external entities in the healthcare system. Separate entities are isolated, but this strategy maintains a consistent and transparent flow in a data-privacy-preserving manner.

Users have complete ownership over their data, which is saved on peer-to-peer blockchain networks, guaranteeing perfect data security and privacy. The basic issues with cutting-edge home applications were investigated in [24], which also offered a safe and effective smart home architecture to solve these problems. The suggested system also satisfies the security requirements for communication security, scalability, system effectiveness, and defensive performance against diverse threats.

There are three methods for revealing the location of blockchain addresses, all of which have the potential to compromise privacy. As a result, the writers created a novel blockchain approach to secure the works position while increasing the job success rate [25].

"Regarding research networks-of-networks, proposed research [26] concentrated on establishing a hierarchical way to inherit the privacy-preserving features and retain blockchain adoption services. The authors created a system that integrates a hierarchical consensus algorithm, blockchain-based model distribution, and model learning".

When a user group is geographically proximate, the proposed work [27] looked at the privacy risk created by attackers who utilize data mining algorithms to trespass on a user's privacy. The authors suggested using the black-box module to create a smart contract. This module allows frequently executing power trading activities on demand to maintain privacy in design objectives.

Proposed research work [28] investigated the weaknesses in two existing private information systems and proposed an "Ethereum—base" location privacy preservation solution. Instead of using a third-party anonymizing server, the proposed approach follows the k-anonymity security principle.

Conflux consensus protocol encodes two alternative key generation algorithms to defend against the active liveness assault [29]. The conservative approach ensures progress

toward consensus, whereas the optimal approach ensures speedy confirmation. Conflux is a decentralized, scalable, high-bandwidth blockchain with rapid verification. It combines these two strategies into a single integrated consensus process using a ground-breaking adaptive weight mechanism.

To tackle the double-spending attack, proposed research [30] developed the MSP (Multistage Secured Pool) framework, which allows the pool to authenticate transactions. The proposed structure contains four steps to counter this attack: (1) recognition, (2) verification, (3) transmission, and (4) dissemination, all steps in the process. Begum et al. [31] also propose ways to prevent double spending assaults after establishing the attack restriction.

To examine the 51 percent offensive, proposed research [32] focused on crypto-coins with low hashing power, identifying the weakness in the consensus process that permits this attack. The authors have defined the hash rate difficulty and present five security approaches for avoiding a 51 percent attack. Recent work to mitigate the 51 percent assault includes defensive mining and implementing a "Perm point" finality arbitration system to reduce chain reconfiguration [33].

Proposed research [34] advised leveraging public-key encryption in distributed ledger technology to identify entities to mitigate a vital security attack. This strategy maintains the integrity of the blockchain network. The topic of group key management is discussed to safeguard group communication and network confidentiality.

An overview of security mistakes, transaction implementation, and a secure operation methodology policy is given in [35]. One scheme is deployed to improve the influence of threats/error and also improve the networks solidity processing, which is a homomorphic cryptosystem, ring signature, and numerous other security features.

The vulnerability of self-mining has offered a strategy to mitigate this attack [36]. To resist the assault, the creators use an accurate removal method to design a truth state notation for units, the self-removal fork, and assign self-authorization height to each contract. A thorough analysis of self-mining assaults and associated countermeasure strategies is conducted in [37].

The "DA" insider attack on the RPL IoT network was handled in proposed research. The authors developed a technique to prevent this assault by conducting trials with the Contiki tool, a low-power built a tool for resource-restricted systems [38].

Because of its distributed nature, blockchain is a good fit for the Internet of Things [39,40]. It is projected to play a crucial role in the administration, control, and security of the most critical apparatus. It will offer new security solutions for numerous aspects of the Internet of Things, including the security of the network environment, hardware management, and data confidentiality, among others. Blockchain-based node authentication is currently a popular topic in the study.

The proposed work [41] uses edge nodes to link IoT nodes that cannot install blockchain software to the blockchain network, manage and authenticate node identification, and allot cloud resources using the concept of trust. However, they do not specifically explain how to do so.

In a broad sense, private chains include both consortium blockchain and private blockchain. Any individual, team, or organization that shares a ledger is considered a public blockchain. Any business or individual may participate in the agreement process as long as the chain recipient can upload transactions on it and the blockchain technology can properly authenticate it.

The public blockchain is the first and most widely used blockchain that is thought to be completely decentralized. A "consortium blockchain" is one in which the agreement process is controlled by selecting pre-selected nodes. Private blockchain refers to utilizing blockchain solely for accounting reasons and is not accessible to the general public. This sort of blockchain is referred to as "partially decentralized." Its target may be a business or a person who only has written access to the blockchain and perhaps only limited access to

the rest of the world. To tackle more problems, blockchain is projected to be merged with smart artificial intelligence and intelligent algorithms [42–50].

Blockchain is a distributed database system with immutable ledgers vulnerable to attacks from unscrupulous users. Despite the fact that the advantages of blockchain have just been leveraged since the first virtual currency to the present shared ledger, the new technology must rely on cryptography for security. Although various papers focus on the weaknesses and security of blockchain, there is a lack of a thorough and systematic evaluation from both the application and technical perspectives [51].

Although blockchain promises to make information exchange across industrial partners easier, decision-makers must understand their conditions in their specific business contexts. Another practical challenge in developing a blockchain-based system is scalability [52]. It may be utilized only if blockchain is appropriate and provides better security for attaining more business benefits.

Several publications recommended ensuring an ad hoc on-demand distance vector, a reliable routing protocol based on preceding encoding that can withstand specific routing assaults while retaining the dependable and recognition of identity, as indicated in [53,54]. Other authors proposed a safe routing design that is low-maintenance, energy-conscious, preserves a trusted environment, and separates misbehaving nodes. The authors presented an intrusion-tolerant routing method for wireless sensor networks. A rogue node can compromise a few nodes in a constrained area but cannot take down a significant chunk of the network.

Bitcoin and other cryptocurrencies are boosting interest in blockchain. Two classic centralized institutions, governments and banks are just starting to show interest in blockchain technology.

The term distributed ledger technology is starting to gain traction in the world of cryptocurrencies. However, a lot of people mistake blockchain for distributed ledgers and vice versa. A distributed ledger with a particular set of technological foundations is known as a blockchain. After an agreement on all the records, the blockchain generates an immutable ledger of data maintained by a decentralized network. A distributed ledger is a database that exists in various places or among numerous users. However, the majority of businesses continue to employ centralized databases with permanent locations. A distributed ledger is decentralized, as opposed to a centralized database, which helps to do away with the requirement for a central authority or middleman to process, validate, or authenticate transactions [55].

In the next section, there is a proposed security system but before starting our work we need to understand blockchain technology and digital twins for blockchain technology, which means how digital products work in the blockchain environment. A blockchain is an ever-growing collection of documents that are connected by cryptography and contain transaction information, a timestamp, and a cryptographic hash of the block before it. Blockchain data are kept in a distributed ledger that cannot be modified until all subsequent blocks have been updated with the new information. We now focus on the transaction in the digital twin-based blockchain and how are data packets transferred. In order to handle new problems and difficulties, DTs have been embraced in the "fourth industrial revolution." In light of the fact that blockchain transaction blocks—the fundamental unit for storing and disseminating a DT's public key, private key, timestamp, and product life cycle—are the main focus of this section, they are the primary subject of discussion. The private key is a key that each user must have in order to access their cryptocurrency; the public key is a shared key that is exclusive to all network participants [56].

**Table 1.** Comparison table of previous studies.

| Paper | Security | Efficiency | Reliability | Decentralized | Drawbacks |
|---|---|---|---|---|---|
| [7] | Moderate | Moderate | Yes | No | Integrity is possible in this system |
| [14] | Low | Moderate | No | Non-centralized | The verification process is low |
| [17] | Moderate | Moderate | No | Yes | More chances of security attacks |
| [25] | Moderate | Low | No | Yes | An unknown person can attack transmission |
| [28] | Low | Low | No | Yes | Weakness of private information |
| [29] | High | Moderate | yes | Yes | Cannot bear the fault |
| [30] | Low | Moderate | No | Yes | The complication in data transactions |
| [32] | Moderate | Low | No | Yes | Low hashing power |
| [35] | High | Moderate | No | No | No assurance of privacy of data |
| [38] | Low | Moderate | No | Non-centralized | The verification process is low |
| [41] | Low | Low | No | Client-server architecture | Not secure, any node data can be dropped during transmission. |
| [42] | Moderate | Moderate | yes | Peer to Peer | Not secure for private data |
| [48] | High | Moderate | No | No | Privacy assault can be initialized, but it is unreliable for large transactions. |
| [51] | High | Low | No | Yes | lack of reliability and systematic evaluation |
| [57] | Moderate | High | Yes | Yes | Need confidentiality |

Throughout the product life cycle, data sharing is done securely and safely using a key pair. An induction time for a transaction is specified by the timestamp. Data pertaining to the life cycle of a product are linked to the company that manufactures it. According to a blockchain transaction, any modification to the DT will be documented using the public key, the private key, a different timestamp, and the pertinent life cycle information [57].

## 4. Blockchain Technology for Security

According to the procedure, the blockchain contract record is turned into a sensor information record. Each datum block in the blockchain system comprises two components: a header and then a body, which are generated every 10 min. Many pieces of information are in the header block, including the current block number, previous block hash value, current timestamp, random-value (nonce), current block hash value, and Merkle tree.

Most of the sensor data are included in the units or blocks content. Every sensor data point is continuously stored in the data block of the research system, where the operator can access it. Each element of the sensor data acquired is digitally signed by the Merkle tree in the block, ensuring that there are no repetitions of the sensor information. Once the system has received all the sensor data, "Merkle-root values" are created using the Merkle-tree hash algorithm in the block's header [58]. Blockchain architecture is shown in the following figure.

A vital component of the blockchain system is the untranslatability of the timestamp and sensor data. In the contemporary digital age, the timestamp serves as a digitalized postmark and is an essential security component. The timestamp is used to offer exact "proof of time" and certification of every record modification or transaction content since it serves as the time stamp for every electronic file or transaction. The electronic timestamp indicates a message or document's creation date and time by assigning a specific time.

Even if a certificate is terminated or cancelled after a timestamp is created, it still has the nonrepudiation feature.

The timestamp and sensor data cannot be changed, which is an essential feature of the blockchain system. In today's world, the timestamp serves as a digitalized postmark and is an essential security element. The timestamp is used to validate any record modification or transaction content since the moment of stamping for any electronic file or transaction, which provides precise "proof of time." In order to establish when communication or a document was made, the electronic timestamp lends a precise time to it. The nonrepudiation functionality remains active even if a certificate is revoked or canceled after a timestamp is generated.

The central node idea is still there in the dispersed network design, and many central node conceptions may exist. Each primary node is connected to other auxiliary nodes. Decentralized network architecture is one in which there is not a single central node. The content of every node is identical. These two network designs are combined in our system. A distributed network design exists by adding up to the decentralized network structure for data consistency, which boosts the sensor's information transmission efficiency.

The integrity of a blockchain is due to its novel use of cryptography and the proof-of-work approach. On the other hand, blockchains secure themselves in another way: they are dispersed. A blockchain uses a peer-to-peer infrastructure that anybody can join instead of a central entity administering the chain (assuming the blockchain is public). Anyone who joins the "Ethereum blockchain" becomes a node and receives the entire blockchain. This node can then utilize the blockchain copy to verify that no one is in order.

### 4.1. Cryptography Technology Application

All blockchain transaction information is eternally included in the records block for other questions. For many other questions, each blockchain user's blockchain revenue and spending are permanently implanted in the data block. Each blockchain user's user node stores the transactional data in these data blocks; these nodes collaborate to build a distributed ledger technology and database system. The database's overall operation is unaffected by the loss of information inside one node because other competent nodes control the database.

The importance of hash functions in blockchain systems is vital. Along with the original content or contract history, the data in the ledger also contain the document's cryptographic hash values. Put another way, the document is numerically encrypted to a specified length and stored in the blockchain as a string of letters and numbers. The following benefits of using the hash function while compiling blockchain records are offered.

The system cannot erase the impacts of obtaining the needed data values from the output result values since the hashing technique of calculating data is typically one-way. The input of the beginning data (256 bytes) is followed by production of 256 bytes in the first block, application of Merkel's transformation, and so on to the last block. A 256-byte hash value is a result. Merkel's architecture employs the protect "hash algorithm SHA256," a block at a time holding 512 bytes.

For instance, the system uses two distinct input values separated by a single byte. The outcome is significantly different when the hash function is used to compute the output estimate for the two input variables. The output lengths are often the same when two SHA256 hash algorithms provide source material of various lengths. Any amount of message may be compressed using the earlier algorithm to provide a fixed-length message digest.

The hash function is a vital component of the blockchain system, enabling numerous practical encoding scenarios. Along with the hash technique, asymmetric encryption algorithms such as the elliptic curve encryption algorithm encrypt transactions on the blockchain. With this method, material encrypted with one key can only be decoded with another mathematically corresponding key. A set of keys consists of public and unpublished

or secret keys. A private key is the account's password or the owner's signature, but a public key is comparable to a bank account. The blockchain considers a transaction to be efficient when the whole transaction's encryption key signature is a certificate authority signing that can be verified using the transaction initiator's public key. According to the method, the secret key cannot be deduced from the public key, while the public key may be acquired through encryption.

Figure 2 depicts the "ECC" technique used in a blockchain-based system and a sample asymmetrical data encryption. An operating system sends a private key, an arbitrary 256-bit value, to the blockchain-based system. The total number of private keys is 2256, making it difficult to crack them [59].

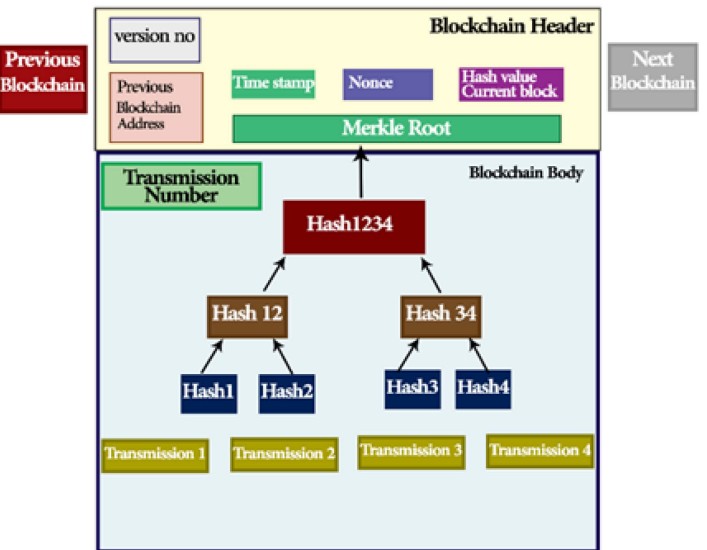

**Figure 2.** Structure of blockchain.

Symmetric key encryption, asymmetric key encryption, and hash algorithms are the three most common cryptographic approaches. The hash algorithm is usually used as the foundation of the blockchain technique. Using fingerprints is one of the various approaches for ensuring the integrity of files. If the user wants to guarantee that the substance of his document cannot be modified, he can stamp his fingerprints at the bottom of the document, preventing others from altering the content or creating a false document because fingerprints cannot be forged. The user can check identification on file with the fingerprint to guarantee that the document has not been altered. If the results of the comparison are different, this document is unique [60].

The system typically handles the data as a document file. Although fingerprinting is also unique, when you calculate the message using the hash function, you have a distinct message digest that you may associate with a fingerprint. The statement, password hash algorithm, and message digest are shown in Figure 3.

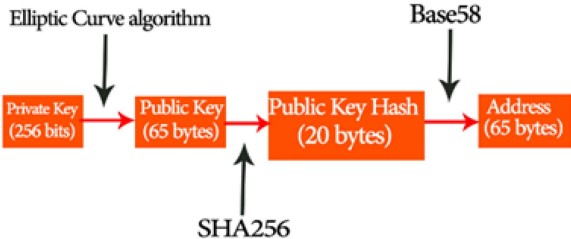

**Figure 3.** Blockchain asymmetric encryption technology.

Despite the fact that these communications are treated as documents and message digests as fingerprints, there are some discrepancies. The most important thing is to keep modifiers out of the message digests. The hash function can be used to re-input the actual content of the message whenever the system wants to check whether the weather decoder has been updated. Comparing the final message digest to the prior message digest is possible. The file check integrity schematic is shown in Figure 4.

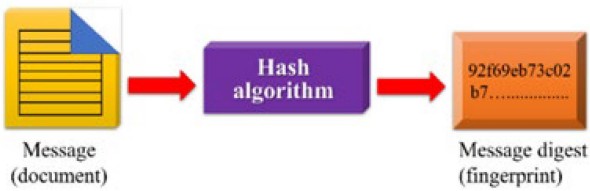

**Figure 4.** The communication and condensation.

Despite the fact that message digests and these messages are viewed as documents and fingerprints, respectively. The message digests must be shielded from modifiers, which is crucial. The calculation of the hash function allows the system to re-input the original message content when determining if the message digest has been altered. The final message digest and the prior message digest can be compared. The file check integrity schematic is shown in Figure 5.

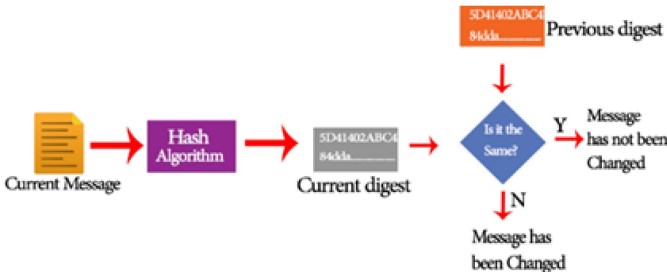

**Figure 5.** A schematic of file check integrity.

### 4.2. Structure of Blockchain

Organizations and corporations can use blockchain infrastructure for the specified objectives:

**Cost Saving**: A lot of money is spent keeping centrally maintained databases, such as those used by banks and other governmental agencies, up to date and secure from cybercrime and other types of corruption.

**Data Retrospective**: You can always check the history of any transaction within a blockchain architecture. This is an ever-growing archive, as opposed to a centralized database that is more like a single data snapshot.

**Security and Validity of Data**: It is difficult to interfere with data once published, owing to the nature of the blockchain. It takes time to validate records since they are validated in each isolated network rather than employing a pool of computational resources. This means the system must choose between efficiency, data protection, and validity [61,62].

### 4.3. BCT Architecture Types

Three different blockchain setups exist.

**Architecture for Public Blockchain:**

A public blockchain architecture allows anybody who wants to join to examine the data and utilizes the system (e.g., Bitcoin, Ethereum, and Litecoin blockchain systems are shared).

**Architecture for Private Blockchain:**

Private blockchain architecture differs from public blockchain architecture in that only users from a single company or users who have been asked to join and have been authorized may administer a private blockchain system.

**Architecture for Consortium Blockchain:**

A few businesses might come together to form the blockchain consortium's architecture. In a consortium, the first selected users establish and control procedures. The table below provides a detailed comparison of these three blockchain technologies. Blockchain, as previously mentioned, is a distributed journal in which each stakeholder has a local copy. The system may be more centralized or decentralized, depending on the kind of blockchain architecture used and the environment in which it is used. This concerns the blockchain's design and who controls the ledger. Due to its administration by a single business and increased anonymity, a smart contract is more centralized. On the other hand, a public blockchain is decentralized since it is an open-ended system [63]. Figure 6 illustrate BCT configuration.

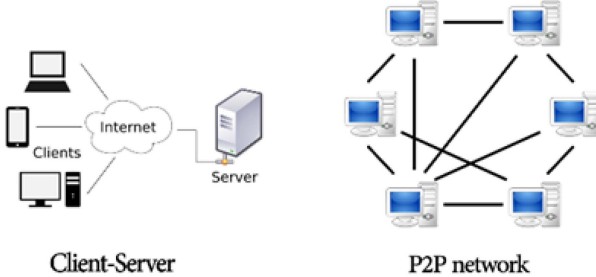

**Figure 6.** BCT Configuration.

In a public blockchain, everyone may participate in the agreement process and can see all records. This is ineffective, however, since it takes a while for the blockchain architecture to accept each new record.

A public blockchain's time for each transaction is low-eco compared to private blockchain architecture since it uses much computational power [64]. Below is the comparison of BCT types in Table 2.

**Table 2.** Evaluation of BCT types.

| Property | Public Block Chain | Consortium Block Chain | Private Block Chain |
|---|---|---|---|
| Consensus Determination | All miners | Selected set of nodes | Within one organization |
| Red permission | Public | Public or Restricted | Public or Restricted |
| Immutability | Impossible to temper | Could be temper | Could be temper |
| Efficiency | LOW | High | High |
| Centralization | No | Partial | yes |
| Consensus Process | Permission-less | Needs permission | Needs permission |

*4.4. Blockchain-Based Technology Is Being Planned for WSNs*

Every block transmits non-rewritable data, ensuring an extraordinarily high level of security. When it comes to secret WSNs, this security is beneficial. A new WSN is being constructed using the most recent blockchain technology [65].

Budget constraints or operational challenges can stymie the development of a WSN. The problems normally begin in local areas and progressively spread to larger areas. The most appropriate method for reporting these issues is to use the current popular blockchain methodology for Bitcoin.

Several sensor devices are linked to each significant node. A string of numbers indicating the chronological sequence in which the blockchain linkages were created is also included in each node. In addition to its sensor data, each blockchain also gathers measurement data from the sensor nodes of other blocks. The central node receives the same sensor data as the new block whenever a new block is formed for each node block. The approach proposed in this study strengthens the security of the interconnecting block methodology by evaluating whichever block to link first in chronological order. Before connecting successfully, authentication using the hash function and password keys approach must be used. Furthermore, no efficiency issue with Bitcoin mining has been resolved. Each primary node is connected to several sensor gadgets. Each individual node also includes a series of numbers showing the order in which the blockchain links were established. Each blockchain collects measurement data from the sensor nodes of other blocks in addition to its sensor data. Each time a new block is formed for a node block, the central node also receives the new block's sensor data. By analyzing which block to link first in chronological sequence, the technique suggested in this research increases the security of the interconnecting block methodology. Before connecting successfully, authentication using the hash function and password keys approach must be used. Furthermore, no efficiency issue with Bitcoin mining has been resolved. The current blockchain node connection is shown in Figure 7.

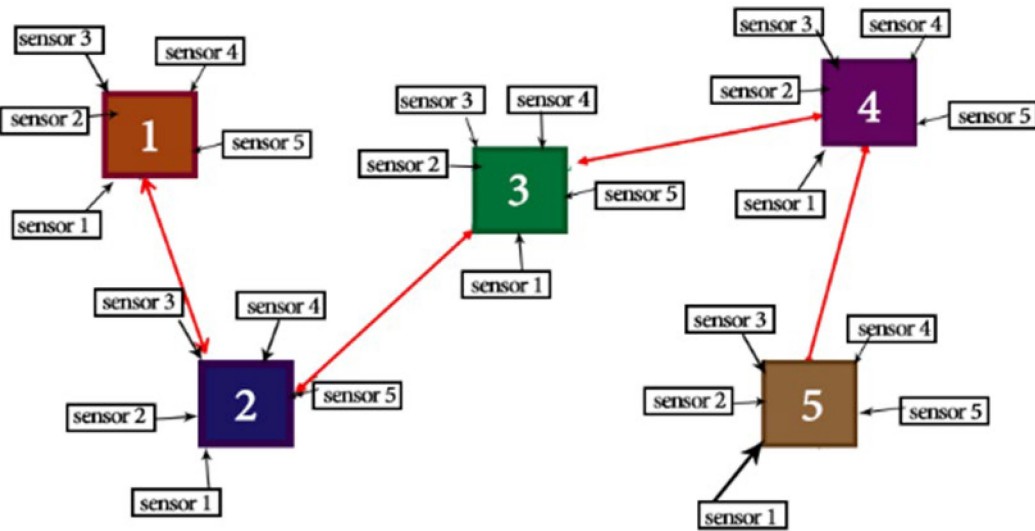

**Figure 7.** Incorporation of a blockchain approach into the structure of a WSN.

As a result, each node keeps track of sensor information for itself and other nodes, and no single node serves as the main center point node, proving the use of decentralization. The P2P network connects all blockchain nodes, and the team uses the safest encryption possible based on network cryptography computations. The incorporation of the blockchain technology method in a WSNs structure is depicted in Figure 8 as shown below.

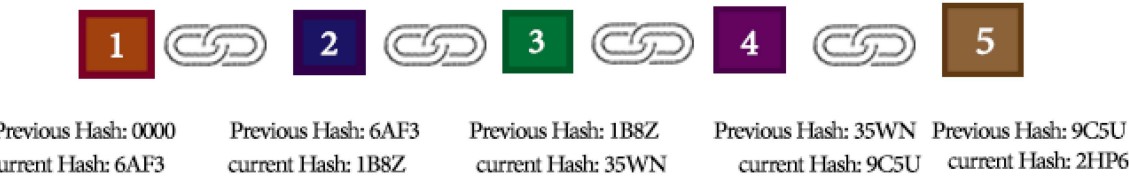

**Figure 8.** In a typical WSN, a blockchain planning instance for preferential nodes.

In addition, the ensemble's renewing functions have been expanded, allowing the blockchain to renew links. Although the order in which the linkages are formed may differ,

the accuracy of the measurement data is assured. Every 30 min, the system's starting condition triggers a blockchain reset.

The asymmetric cryptographic method connects the node orders indicated in the figure, which shortens the mining process and boosts performance effectiveness. Due to the system's ability to apply encryption and cryptography utilizing keys to each freshly produced block, each blockchain is connected to a previous and subsequent opposing side. Each subsequent block successfully connects to the preceding block using the established method of the system. This sets the Bitcoin blockchain strategy apart from competing ones.

Every node has a hash function, both the previous and current blocks. Despite being made up of long word sequences, hash functions are represented in this picture as four single-digit digits. Figure 8 shows a block order with erroneous hash functions, indicating inappropriate linking since the values of the nodes in the order are inconsistent. The system immediately terminates that blockchain's connection and data transmission when it detects such an incorrect connection.

The measurement data for each blockchain node are also guaranteed to remain anonymous. The identical unanimous measurement material is present in every blockchain subblock submitted for all WSNs. The "distributed file system" design dramatically reduces the risk connected to data storage. As a relatively new area of study in WSNs, blockchain applications need software programming of the microcontroller. It is different from the design of conventional WSNs. Our study leverages the Web platform to build the system in addition to the blockchain, which eliminates the need for numerous communication protocol standards and vastly increases efficiency. Data loss due to node failure is not a concern since each block node offers substantial sensory data. Then we will look at how blockchain works, including how it creates a chain and stores and transfers data.

## 5. Results Analysis

As mentioned above, there is a description of blockchain, which is the architecture and how it works. All are described with graphics. Blockchain is a rising security technology in every networking era as this system is about the security of clustering in WSNs. Small clusters contain sensor nodes that transmit the data over the network. Blockchain technology is implemented to ensure integrity, and that data will not change during transmission.

The block is the core component of the blockchain. The message content of each block contains:

- The block number
- A timestamp.
- Measurement information from each sensor.
- The hash value of the previous block.
- The randomized Nonce value.
- When the first block of Genesis is produced, the previous block's hash value is "0".

A block is implemented for the first time. The suggested method is used to determine the "block number, sensor data, timestamp, and the hash value of the preceding block," as illustrated below. The hash value is "No." plus "Data," "Time-stamp," and "PreviousHashValue."

Figure 9 depicts the blockchain network's usual connection. The system immediately locates the new location when a set of information is changed. The Nonce and hash code of the 87th block is changed in the peer B Network, as shown in Figure 10. The data in the peer B network's 87th block differ from the data in previous blocks, according to the system's appraisal of the peer A and peer C networks.

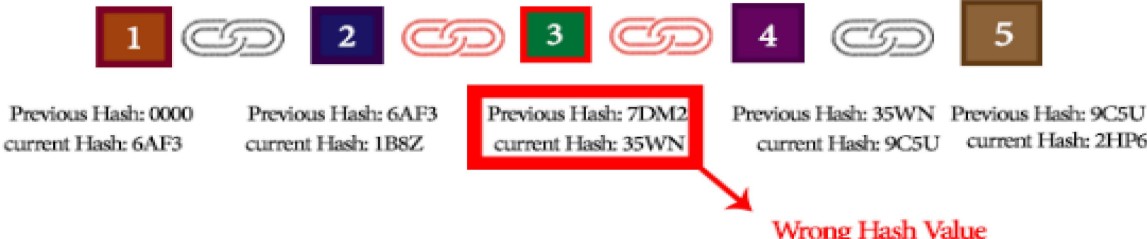

**Figure 9.** Sample blockchain planning for partial nodes in unusual WSNs.

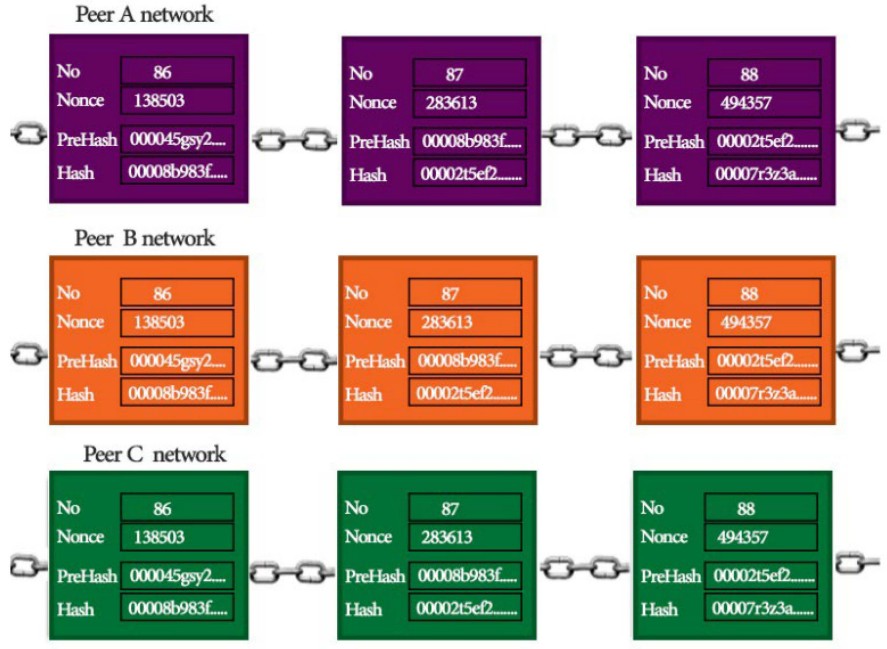

**Figure 10.** In normal circumstances, a blockchain connection is established.

Blockchain technology, a subset of crypto technology, is now gaining popularity among academics and is being used in various industries, including big data, artificial intelligence (AI), and virtual reality (VR).

For better comprehension and viewpoint of this technology, think about using Google Earth as an analog. Although not a groundbreaking innovation, Ajax may be used with other technologies, such as Google Earth, to create something extraordinary. When the blockchain technology is integrated with other technologies like peer-to-peer networking, encryption, and decryption, bitcoin is created.

The connections for blockchains are made after establishing the block data portion style. The original genesis block is created from scratch, and for each step taken, associated programmers are executed. The system must compute the blockchain's length according to the software when the most recent block is established.

In the framework of WSNs access sensor data records, this study compares the blockchain technique and the traditional approach.

The connections for blockchains are made after establishing the block data portion style. The original genesis block is created from scratch, and for each step taken, associated programmers are executed. The system must compute the blockchain's length according to the software when the most recent block is established. In the framework of WSNs access sensor data records, this study compares the blockchain technique and the traditional approach as shown in Figure 11 in the peer B network the Nonce and hash values of the 87th block are tampered. When the system compares peer A and peer C networks the system finds that the data in the 87th block of the peer B network are different from those of other blocks.

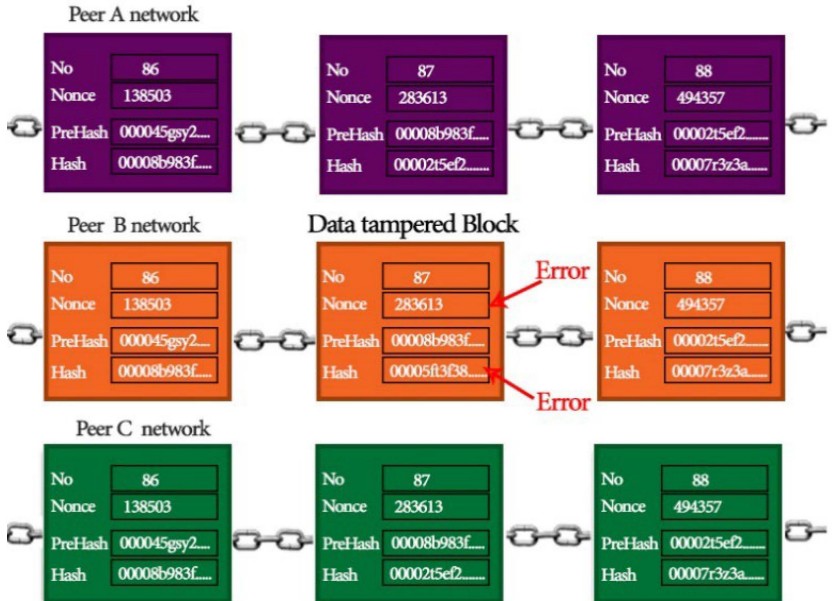

**Figure 11.** After being tampered with, the 87th block data.

Figure 12 shows how the traditional (endpoint security method) and blockchain approaches compare regarding the number of information records that can be processed every half hour. The chart indicates that the blockchain-based technique has fewer records than the old way. Even so, it is a minor difference, and its show execution is virtually as good as other systems.

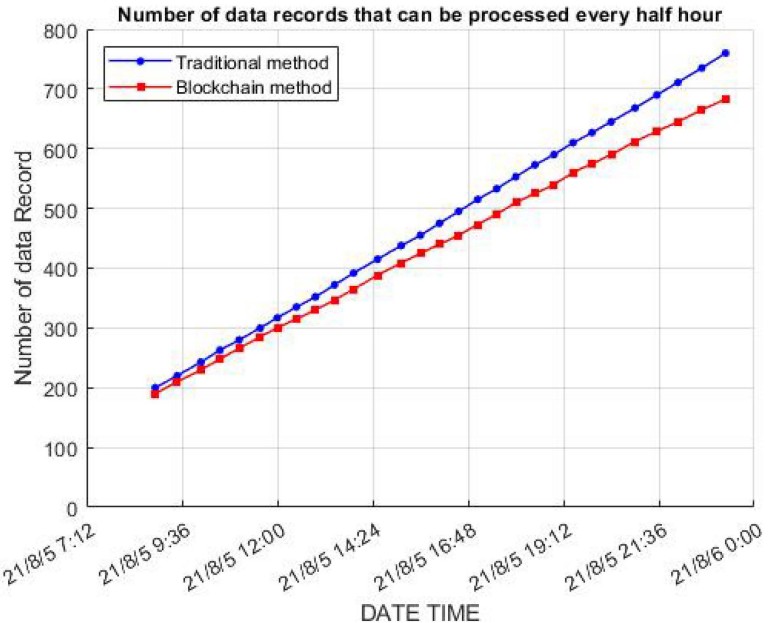

**Figure 12.** Traditional and blockchain-based methods are compared.

This difference is nevertheless acceptable to operators. The main goal of our study was to increase the security of wireless sensor networks.

The first block, or the genesis blockchain, has an index key value of "0," whereas the value of the subsequent block is "1." It is crucial to check whether the previous hash function was correctly replicated before adding a new block. Figure 13 depicts the proposed system, which is made up of several programmed files, each representing a subsystem.

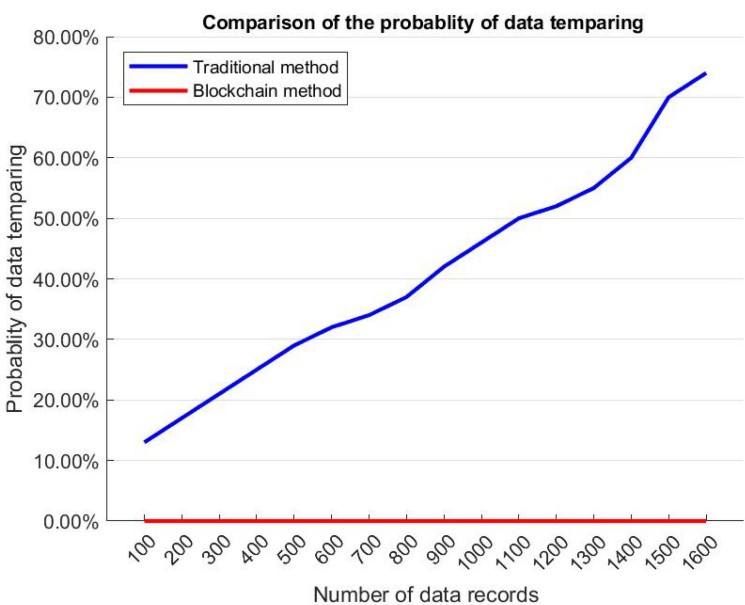

**Figure 13.** The probability of data manipulation is compared.

Figure 12 shows a difference in sensing data transmission evaluation. The overall efficiency of the system improves because the system uses the Web platform to simplify the data format of various agreements when the data exceed the number. Microcontrollers, on the other hand, favor built-in memory since we need software to address numerous data-transfer problems, such as converting data formats or switching protocol platforms.

Figure 13 illustrates how a system's reliance on conventional or blockchain technology affects the likelihood of data manipulation. The recommended method uses the conventional wireless sensor network technique when the quantity of sensed data is significant, and the data supplied by it are more likely to be tampered with. One advantage of the blockchain method application is that when the suggested system with the blockchain method processes a sizable quantity of data under the same data transmission circumstances, the data sent by the wireless sensor network cannot be covertly manipulated.

As depicted in Figure 14, the amount of registration phase increases parallel to authentication phases, which are increased daily. In networking, the most important thing is the security of their work or data.

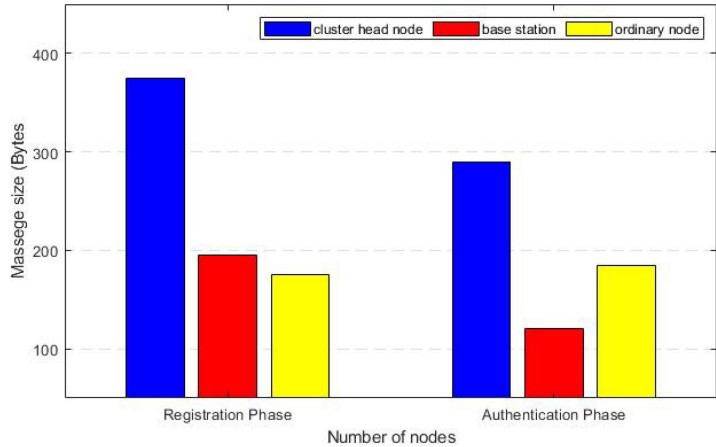

**Figure 14.** Comparison of message sizes transmitted by different nodes.

There are several applications of blockchain, but it is seen that it is primarily used for digital currencies. Figure 15 highlights the taxonomy of blockchain platforms. It is to be

noted that we are not endorsing any of the mentioned platforms. Moreover, it should not be interpreted as the catalogue of the most prevalent platforms. The above figure presents various platforms where blockchain is used for the security of the network. As mentioned earlier, blockchain is more secure but due to high security it may work slowly for this different protocol used or some algorithm that solved the Byzantine problem.

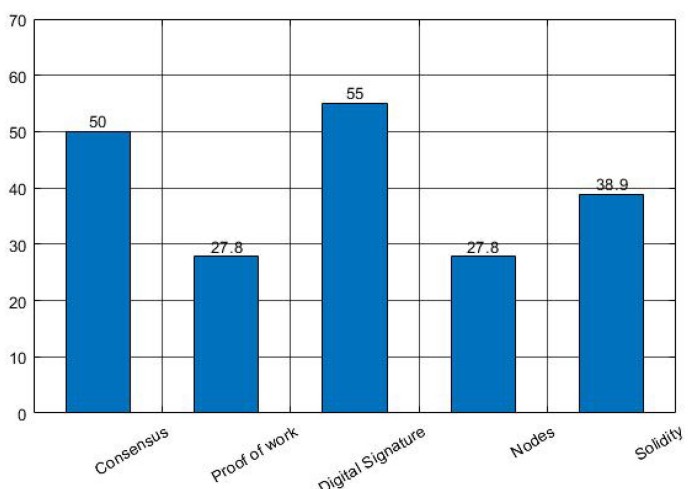

**Figure 15.** Various platforms in which blockchain is being used.

## 6. Conclusions

We work in the broader field, the Internet of Things (IoT); in this case, we present blockchain technology for the security of WSNs. We talked about the wireless sensor networks (WSNs); in this network, we chose a cluster with a Cluster Head Base Station and Nodes. We also discussed the LEACH protocol, which is used for the energy efficiency of the network, and some drawbacks of the LEACH protocol. As everyone knows, blockchain is an emerging technology for the security of every network in the Internet of Things, e.g., smart home, smart cities, healthcare systems, agriculture, food packaging, structural monitoring, remediation of groundwater, accident avoidance, wildlife monitoring, and home automation system.

To link a blockchain, cryptography processing and public key decryption are needed. Additionally, in order to transmit data when constructing a new blockchain, the link's earlier and later sequences must be confirmed. Therefore, various problems must be resolved in order to accomplish immediate data updating and storage.

The hash function and encryption key computation are necessary for the suggested blockchain-based method, and they cannot be avoided. However, the efficiency of data transport decreases as the amount of data increases since more calculations take longer to complete. Large-scale calculations will therefore lengthen the data processing time.

Our team is working to address the aforementioned issues using techniques like the blockchains' reset mechanism, which will always update data transmission to the most recent state. The application of a system control mechanism and a streamlined hash function computation procedure is another technique that could be helpful. Additionally, symmetric encryption will replace asymmetric encryption to streamline blockchain security.

Several advantages will result from integrating blockchain technology into numerous information applications. Blockchain technology effectively reduces the danger of data tampering when the data are transmitted by using a decentralized and universal consensus process to maintain the integrity of the data. The blockchain-based ledger that was first utilized in banking can be compared to the sensory database of the WSN. Based on blockchain technology, each block contains an unforgeable timestamp and a more restricted record database.

As a result, we assumed that blockchain is more secure than other security techniques. According to the result analysis mentioned in Chapter 4, the reliability and the rate of data tampering with blockchain are low, which means there is a lower percentage of malicious attacks.

**Author Contributions:** Conceptualization, A.R. and S.A. (Saima Abdullah); methodology, A.R. and M.F.; software, M.W.I.; validation, S.A. (Saqib Ali), M.U.A. and M.W.I.; formal analysis, M.U.A. and M.W.I.; investigation, S.A. (Saima Abdullah); resources, K.A.A.; data curation, A.R.; writing—original draft preparation, A.R. and S.A. (Saima Abdullah); supervision, M.F.; project administration, M.U.A.; funding acquisition, K.A.A. All authors have read and agreed to the published version of the manuscript.

**Funding:** This research was funded by Deanship of Scientific Research at Umm Al-Qura University grant number 22UQU4310108DSR03.

**Institutional Review Board Statement:** Not applicable.

**Informed Consent Statement:** Not applicable.

**Data Availability Statement:** All data used in research is mentioned in the paper.

**Acknowledgments:** The authors would like to thank the Deanship of Scientific Research at Umm Al-Qura University for supporting this work by Grant Code: (22UQU4310108DSR03).

**Conflicts of Interest:** The authors declare no conflict of interest.

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
