# Peer review of "Ensuring Security and Energy Efficiency of Wireless Sensor Network by Using Blockchain"

_applsci, doi:10.3390/app122110794_

Round 1
Reviewer 1 Report
The manuscript presents an interesting study using the blockchain approach to ensure the security and energy efficiency of the wireless sensor network. The manuscript is designed well, and it provides useful outcomes. The authors have made a good effort to explore this idea. The manuscript has been written and designed well. I recommend making the following suggestions to improve the manuscript
1- The abstract is incomplete, as it did not present the results and study outcomes and did not provide a conclusion and the study's contribution to ensuring the security and energy efficiency of wireless sensor networks.
2- I have not observed any discussions comparing this study with other studies.
3-The authors should discuss their findings compared to previous studies and the limitations of their study.
4-The conclusions section needs to focus on the outcomes of the study, its limitations, and future directions.
Author Response
We would like to thank the Reviewers for taking the time and effort necessary to review the manuscript. We sincerely appreciate all valuable comments and suggestions, which helped us to improve the quality of the manuscript.
please see the attached document for detailed response against valued comments .

Reviewer 2 Report
The overall quality is acceptable. The literature aspects can be strengthened.
1) Comparisons between non-Blockchain and Blockchain approaches
SparkNoC: an energy-efficiency FPGA-based accelerator using optimized lightweight CNN for edge computing. Journal of Systems Architecture . (2021), 115, 101991.
2) Making digital twin for Blockchain and make comparisons
Evaluation of Production of Digital Twins Based on Blockchain Technology. Electronics (2022), 11(8), 1268.
Please strengthen research contributions. Please double check grammar and proofreading.
Author Response

(The authors gave the same response as above.)

Reviewer 3 Report
Comments:
1. Page 2, line 50. There should be a ‘the’ before system.
2. Page 2, line 69. Please check the rationality of applying the word ‘Inconsequential’.
3. Page 3, line 71. Please give the reference for the ‘existing issue’.
4. Some References are missing, such as [1], [2], [5], and [6].
5. Page 3, line 108. Introducing block chain technology is not a contribution.
6. Page 3, line 182. Check the format of the first sentence.
7. Page 6, line 244. What is the caption of the Table 1?
8. Page 9, line 389; page 10, line 399. The similar content has been repeatedly shown.
9. Page 11, line 446. Check the grammar of ‘is like’.
10. Page 14, line 522. Check the order of the table 1.
11. Page14, figure 7. ‘Each blockchain collects measurement data from the sensor nodes of other blocks in addition to its sensor data’. Is that mean it will cost more energy to collect data to ensure the data security?
12. Most of the figures are not a high resolution.
13. Page 16, 612; Page 17, line 629. The similar content repeats again.
Author Response
We would like to thank the respected Reviewer for taking the time and effort necessary to review the manuscript. We sincerely appreciate all valuable comments and suggestions, which helped us to improve the quality of the manuscript.
please see the attached document for detailed response against valued comments .

Reviewer 4 Report
In this paper, privacy and security issues in the IoT system are examined and a Blockchain technology-based solution is proposed for security issues. The contribution of the study to today's technology level is not clear. It is a generic paper that collects well-known information about IoT and Blockchain. Above all, the paper contains similarities with the study titled "Employing Blockchain Technology to Strengthen Security of Wireless Sensor Networks" published in ACCESS magazine in 2021 (S. -J. Hsiao and W. -T. Sung, "Employing Blockchain Technology to Strengthen Security of Wireless Sensor Networks," in IEEE Access, vol. 9, pp. 72326-72341, 2021, doi: 10.1109/ACCESS.2021.3079708). Authors should fully reveal aspects that differ from this study. The organization of the paper must be reconsidered throughout; also, there are major concerns, which need to be addressed.
The abstract is not well written. The authors need to rewrite this section and should focus on the novelty and comparative analysis performed in the paper.
At the end of the introduction section, there should be a paragraph about the novelty of the proposed and performed work (Ex: In this paper/study ...).
Please introduce every acronym before using it in the text. The first time you use the term, put the acronym in parentheses after the full term. Thereafter, you can stick to using the acronym. For example:
On 1st page line 41, IoT acronym first appears here, authors should firstly introduce Internet of things, then you can use the term thereafter. But, on 1st page line 44, author use again full term “Internet of Things”. On 2nd page line 47, authors put the acronym. Please follow the academic writing standards.
Please use formal academic writing standards. Some writing problems listed below:
On 1st page, line 27, authors states as “we’ll check”. This is not often used in formal writing. It should be corrected as “we will check”
On 2nd page, line 52, 53, 98, On 3rd page line 119, 125 etc. authors use ampersand (&) instead of and. An & is a typographical symbol that is rarely used in formal writing. It is read aloud as the word and and is used as a substitute for that word in informal writing and in the names of products or businesses. Please use “and” instead of “&”
On page 3, line 108, authors state that “Our Contribution: This study paper makes many significant contributions” This sentence has no scientific meaning. What are the major contributions of paper? This statement must be concretely justified.
On page 20, line 685, authors states that "the reliability and the rate of data tampering with Blockchain is low, which means there is less percentage of malicious attacks." How do you prove this statement?
Some figures (Fig.2, Fig 3, Fig 5, Fig 7, Fig 9, Fig 14) are not clear, and their quality needs to be improved. Also, some figures (Fig. 12, Fig 13) should be replotted, and clean, legible drawings should be presented.
Figures, Tables, and some captions are not well formatted. Figures and Tables should be aligned with text.
The conclusion section is not well written. The conclusion section should summarize the overall study. The contribution of the paper to the current literature should be clearly stated comparatively (especially, S. -J. Hsiao and W. -T. Sung, "Employing Blockchain Technology to Strengthen Security of Wireless Sensor Networks," in IEEE Access, vol. 9, pp. 72326-72341, 2021, doi: 10.1109/ACCESS.2021.3079708)
Author Response

(The authors gave the same response as above.)

Round 2
Reviewer 1 Report
The authors have greatly improved the manuscript.
Author Response
We would like to take this opportunity to thank you for the effort and expertise that you contributed towards reviewing the article, without which it would be impossible to maintain the high standards of peer-reviewed journals.

Reviewer 2 Report
Some image quality can be slightly improved. You may accept the paper for science.
Author Response

(The authors gave the same response as above.)

Reviewer 3 Report
1. Page 2, line 54. There should be a ‘the’ before system.
2. Page 2, line 72. Please check the rationality of applying the word ‘Inconsequential’.
3. Some References are missing, such as [1] and [2].
6. Page 6, What is the caption of the Table 1?
7. Page 11, line 446. Check the grammar of ‘is like’.
8. Page14, figure 7. ‘Each blockchain collects measurement data from the sensor nodes of other blocks in addition to its sensor data’. Is that mean it will cost more energy to collect data to ensure the data security?
9. Page 5, check the word "5entralized"
10. Page 3. Introducing block chain technology is not a contribution. Change it if possible.
Author Response

(The authors gave the same response as above.)

Reviewer 4 Report
The paper is technically sound and timely. Firstly, the paper examines privacy and security issues in the IoT system, and proposes Blockchain technology-based solution for security issues. The contribution of the study to today's technology level is not clear. It is a generic paper that collects well-known information about IoT and Blockchain. Above all, the paper contains similarities with the study titled "Employing Blockchain Technology to Strengthen Security of Wireless Sensor Networks" published in ACCESS magazine in 2021 (S. -J. Hsiao and W. -T. Sung, "Employing Blockchain Technology to Strengthen Security of Wireless Sensor Networks," in IEEE Access, vol. 9, pp. 72326-72341, 2021, doi: 10.1109/ACCESS.2021.3079708). Authors should fully reveal aspects that differ from this study.
On the other hand, When the previous and revised versions of the paper are evaluated together, it is seen that the authors make some corrections requested by the referees and do not show the necessary sensitivity in the writing and improving paper in line with the comments. In the revised version of the paper, almost all the comments have been considered and addressed by the authors.
The response to reviewers file is not well-prepared. The changes made by the authors in line with the opinions/suggestions/evaluations of the referees cannot be tracked.
Finally, although I consider the paper as lacking in novelty and insufficient in terms of contribution to the recent literature, this revised version is sufficient, and suitable for the scope and content of the special issue of Applied Science Journal titled Applied and Innovative Computational Intelligence Systems". Before publication, I strongly recommend that authors should improve the paper by demonstrating their difference from the paper entitled "Employing Blockchain Technology to Strengthen Security of Wireless Sensor Networks".
Author Response

(The authors gave the same response as above.)
